# Modulation of Aub–TDRD interactions elucidates piRNA amplification and germplasm formation

Nicholas Vrettos[1], Manolis Maragkakis[2], Panagiotis Alexiou[3], Paraskevi Sgourdou[4], Fadia Ibrahim[1], Daniel Palmieri[1] ![ORCID], Yohei Kirino[5] ![ORCID], Zissimos Mourelatos[1] ![ORCID]

Aub guided by piRNAs ensures genome integrity by cleaving retrotransposons, and genome propagation by trapping mRNAs to form the germplasm that instructs germ cell formation. Arginines at the N-terminus of Aub (Aub–NTRs) interact with Tudor and other Tudor domain–containing proteins (TDRDs). Aub–TDRD interactions suppress active retrotransposons via piRNA amplification and form germplasm via generation of Aub–Tudor ribonucleoproteins. Here, we show that Aub–NTRs are dispensable for primary piRNA biogenesis but essential for piRNA amplification and that their symmetric dimethylation is required for germplasm formation and germ cell specification but largely redundant for piRNA amplification.

## Introduction

PIWI proteins belong to the Argonaute family of RNA binding proteins; they are expressed in the germline of all animals and bind to small RNAs termed (piRNAs) PIWI-interacting RNAs ([1]). A major and ancestral function of PIWI proteins and piRNAs is to suppress retrotransposons and viruses ([1], [2], [3], [4]). The PIWI domain of Argonaute proteins contains an RNAse-H fold that can cleave target RNAs complementary to their bound (guide) small RNA, whereas the PAZ domain binds and protects the 3′ end of the guide RNA ([1]). The N-terminus of PIWI proteins contains arginines (NTRs) that are symmetrically dimethylated (sDMA) by protein arginine methyltransferase 5 (PRMT5) ([5]), known as Capsuleen/Dart5 (Csul) in *Drosophila* ([6], [7]), and bind to Tudor domain–containing proteins (TDRDs) ([8], [9], [10], [11], [12]). Primary piRNAs are derived from long, single-stranded RNAs that are processed on the surface of mitochondria ([1]). PIWI proteins are intimately involved in piRNA biogenesis by using their MID domain to bind the 5′ phosphate of longer piRNA precursors, protecting a ~26- to 30-nucleotide fragment that will give rise to the mature piRNA, and positioning the Zucchini endonuclease to cleave the precursor right downstream of the PIWI footprint ([13], [14], [15]). A second PIWI protein may use the newly created 5′ end of the precursor to generate another phased (trailing) piRNA, and the process may be repeated until the entire precursor RNA is converted to piRNAs ([13], [14], [15], [16]). The initial cut of the piRNA precursor is often generated by piRNA-guided cleavage ([17]).

*Drosophila melanogaster* expresses three PIWI proteins termed Aubergine (Aub), Piwi, and Ago3 ([18], [19], [20], [21]). Most primary piRNAs are derived from piRNA clusters, which contain sequence fragments of retrotransposons, often arranged in an antisense orientation, as a form of molecular memory of past retrotransposon activity ([20]). Piwi-bound piRNAs are imported to the nucleus where Piwi functions in chromatin silencing of nascent transposon transcripts ([1], [22]). In cytoplasmic, perinuclear structures known as nuage, Aub–piRNAs target and cleave transposons, and the piRNA response is amplified by successive rounds of Aub and Ago3 interactions, in a process known as heterotypic ping-pong ([1], [20], [21], [22]). The Krimper (Krimp) TDRD is essential for piRNA amplification by assembling a complex of methylated Aub bound to piRNAs that are antisense to transposons and nonmethylated Ago3 that receives the Aub-generated, cleaved, retrotransposon products to form sense piRNAs. Ago3 is then methylated and presumably released from Krimp ([22], [23], [24]). The DEAD box protein Vasa (Vas) facilitates transfer of cleaved piRNA precursors during heterotypic ping-pong ([25]), whereas homotypic Aub–Aub ping-pong is suppressed by Qin (Kumo) ([26], [27]).

During *Drosophila* oogenesis, germline mRNAs in the form of ribonucleoproteins (mRNPs) assemble at the posterior of the oocyte to form germ granules in a specialized cytoplasmic structure known as germ (pole) plasm. The germplasm is transmitted to the embryo and, its mRNPs are necessary and sufficient to induce the formation of primordial germ cells (PGCs, germ stem cells) from undifferentiated cells ([28], [29]). Genetic studies have identified factors that are critical for germplasm formation and among them are Tudor (Tud), a large protein containing 11 TUD domains ([30], [31]), Aub ([19]) and Csul ([6], [7]). sDMAs in Aub N-terminus, generated by Csul, are required for

[1]Division of Neuropathology, Departments of Pathology and Laboratory Medicine, University of Pennsylvania, Philadelphia, PA, USA  [2]Laboratory of Genetics and Genomics, National Institute on Aging, Intramural Research Program, National Institutes of Health, Baltimore, MD, USA  [3]Central European Institute of Technology, Brno, Czech Republic  [4]Departments of Genetics, Perelman School of Medicine, University of Pennsylvania, Philadelphia, PA, USA  [5]Computational Medicine Center, Sidney Kimmel Medical College, Thomas Jefferson University, Philadelphia, PA, USA

Correspondence: mourelaz@uphs.upenn.edu

germplasm assembly in vivo (5) via interactions with Tud (9, 10). Structural studies have shown that extended TUD domains (eTUD) of Tud specifically recognize sDMAs and surrounding Aub backbone and support a multivalent Aub–Tud interaction (32, 33, 34). Aub-bound piRNAs tether and trap mRNAs to the germplasm in a Tud-dependent manner to form the germline mRNPs that are essential for PGC specification (35) and piRNA inheritance, which will initiate piRNA biogenesis and transposon control in the germline of the offspring (36).

Here, we report that the dual role of Aub in transposon control and germline mRNP formation is orchestrated by Aub–NTRs and their methylation status. We find that Aub–NTRs are dispensable for primary piRNA biogenesis but essential for piRNA amplification and that their symmetric dimethylation is required for germplasm formation and germ cell specification but largely redundant for piRNA amplification.

# Results

## Arginine (R) to lysine (K) mutation in Aub is a new hypomorphic allele

We engineered an *aub* mutant by replacing the four arginine residues (R11, R13, R15, and R17) that are subjected to symmetrical dimethylation, with lysines (RK), and inserted three tandem HA epitopes (3xHA) at the N-terminus. A corresponding wild-type (WT) *aub* rescue construct was also created (Fig 1A). The RK mutation in Aub abolishes direct interactions with Tud (9) and Krimp (24) but not Qin (Kumo) (27). Transgenes were recombined downstream of UASp promoter, and select WT and RK lines were expressed by *nos*-Gal4-VP16 germline–specific driver under a heteroallelic *aub* null background (*QC42/HN2*) (Fig S1A and B). Thus, the only source of Aub in these flies is from the HA-tagged transgene. To examine Aub methylation, we performed anti-HA immunoprecipitations (IP) and probed the immunoprecipitates on Western blots (WBs) with SYM11, an antibody that specifically recognizes sDMAs. As shown in Fig 1B, SYM11 reactivity is lost specifically in *aub^RK^*, consistent with R11, R13, R15, and R17 being the main arginines in Aub that are subject to symmetric methylation. We find equivalent protein levels of Aub, Ago3, Piwi, Tud, and Vas between *aub^WT^* and *aub^RK^* ovaries (Fig 1C). Females from *aub* mutants exhibit low fecundity rates and carry egg chambers with severe axonal defects, and embryos laid by *aub* mothers arrest before gastrulation and never form PGCs (19, 37). We find that Aub^WT^ restores fecundity to normal levels, whereas the egg laying rate in *aub^RK^* is improved compared with *aub* but lags behind that of *aub^WT^* (Fig S1C). In *aub* and other piRNA pathway component mutants, dorsoventral defects appear as a result of inadequate Grk signaling from dorsal follicle cells (38). We find, by immunofluorescent microscopy (IF), that Grk levels and localization pattern appear normal in stage 9-10 *aub^RK^* egg chambers (Fig S1D). More than 60% of embryos laid by *aub^RK^* mothers do not display dorsal appendage abnormalities and are similar to *aub^WT^* (Fig S1E and Table S1). Nevertheless, only 4% of them are able to complete development, but in all cases, the adult offspring are devoid of germline (Table S2, see below). These findings show that unlike the severe germline defects of Aub loss of function, *aub^RK^* flies lay adequate amount of eggs in the first week of maturity, do not display body axis pattern defects, and express piRNA pathway proteins at normal levels. The Aub^RK^ protein itself does not have toxic/gain of function properties,

as it does not affect viability or fertility of flies when expressed alongside endogenous Aub.

## Aub, Tud, and Krimp nuage localization is disrupted in *aub^RK^*

We next examined by IF the localization of relevant PIWI pathway proteins in ovaries from *y w* (wild-type), *aub*, *aub^WT^*, and *aub^RK^* flies. We find that in the absence of endogenous Aub, and in contrast to Aub^WT^, Aub^RK^ does not localize to nuage, irrespective of which promoter drives transgene expression, *nanos* (Figs 1D and S2A) or *maternal α-tubulin* (Fig S1F). In the presence of endogenous Aub, Aub^RK^ localizes to nuage, although in a less granular fashion (Fig S1G), as previously reported (24), mimicking the localization pattern of endogenous Aub seen in nurse cells from *tud* null ovaries (*tud^1^/Df(2R)Pu^rP133^*) (Fig S1H). Similarly, Krimp and Tud do not localize to nuage in *aub^RK^* ovaries, with Krimp aggregating in cytoplasmic bodies (23) and Tud found diffusely in the cytoplasm (Figs 1D and S2A). In *aub*, Ago3 is absent from nuage and concentrates in Krimp bodies (23) (Figs 1D and S2A). Surprisingly, we find that Ago3 persists in nuage structures, although at lower levels than that in *aub^WT^* (Figs 1D and S2A). The nuage localization of Qin and Vas and the nuclear localization of Piwi are unaffected in *aub^RK^* (Figs 1D and S2A). These findings support roles for both Aub–NTRs and Aub RNA–binding capacity in nuage formation. In the presence of endogenous Aub, the RNA binding of Aub^RK^ is sufficient to recruit it to nuage structures nucleated by endogenous Aub, but Aub^RK^ does not properly condensate in granules as it does not associate with Krimp and likely other TDRDs. In the absence of endogenous Aub, Aub^RK^ alone is unable to build nuage structures, and Tud and Krimp are essentially absent from nuage.

## Aub^RK^ is loaded with piRNAs, but piRNA amplification collapses

Next, we examined the piRNAs that are bound to Aub, Ago3, and Piwi in *aub^RK^* and *aub^WT^* ovaries. We performed IPs, extracted bound RNA from equivalent protein amounts, as determined by WB, and analyzed them by denaturing PAGE after 5′ end radiolabeling. We find in *aub^RK^* that all PIWI proteins are loaded with piRNAs (Fig 2A). To further characterize these piRNA populations, we generated cDNA libraries followed by sequence by synthesis. We find that Aub piRNAs and Piwi piRNAs from *aub^RK^* have similar nucleotide lengths (Fig 2B) and display the characteristic 5′ Uridine preference (1U) (39) (Fig 2C) as those from *aub^WT^* indicating that primary piRNA biogenesis is intact and that the RK mutation does not impair piRNA loading to Aub. Ago3 piRNAs derived from sense retrotransposons after heterotypic ping-pong, display a 10th nucleotide Adenosine bias (10A) (20, 21), and are typically trimmed at the 3′ end by the Nibbler (Nib) exonuclease (1, 40). We find that Ago3 piRNAs in *aub^RK^* are longer by one nucleotide than those from *aub^WT^* (Fig 2B), with marked reduction of 10A and increase of 1U (Fig 2C), indicating a drastic reduction of ping-pong amplification. Longer piRNA lengths have also been reported in *Drosophila tud* (8), *Nib* (40), and *papi* orthologues in silkworm (12, 41) and mouse (42). To further analyze the impact of *aub^RK^* in heterotypic ping-pong and piRNA population shaping, we plotted the 5′-5′ position between Aub, Ago3, and Piwi piRNAs in *aub^RK^* versus *aub^WT^*. As shown in Fig 2D, the 5′-5′ position between Aub and Ago3 piRNAs in *aub^WT^* shows a peak at position 10 (blue line), consistent with robust Aub-Ago3 ping-pong, which is abolished in *aub^RK^* (red line). Similar analysis between Ago3 and Piwi reveals the expected phasing signature of ~27 nucleotides (40) of Piwi trailing piRNAs initiating downstream of Ago3

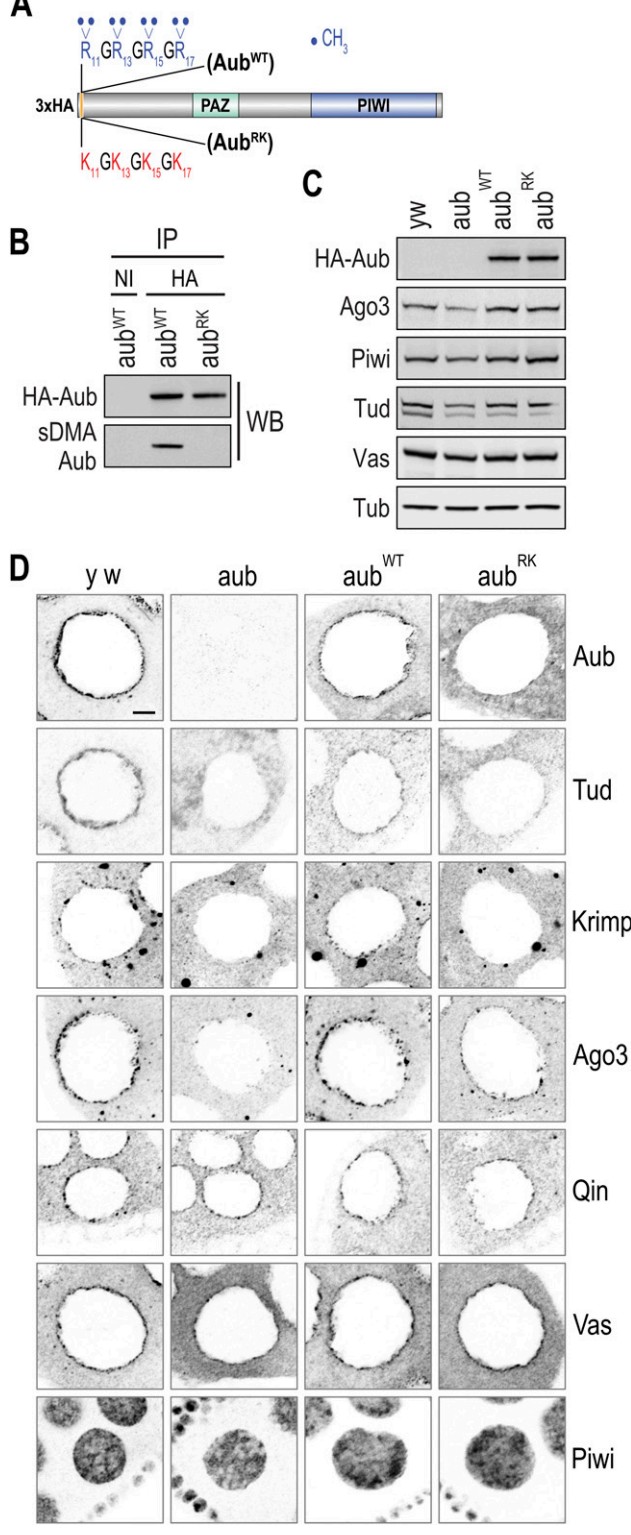

**Figure 1. Aub, Tud, and Krimp do not localize to nuage in $aub^{RK}$.**
**(A)** Schematic representation of wild type (WT) and arginine to lysine (RK) Aub constructs. **(B)** Western blot detection of immunoprecipitated Aub from ovary lysates of indicated genotypes. $aub^{WT} = aub^{QC42/HN2}$; $nos > 3xHA\text{-}aub^{WT}$, $aub^{RK} = aub^{QC42/HN2}$; $nos > 3xHA\text{-}aub^{RK}$. NI, nonimmune serum, sDMA-Aub is detected with SYM11 antibody. **(C)** Western blot analysis in ovary lysates from

cleavage in $aub^{WT}$, which is dramatically reduced in $aub^{RK}$ (Fig 2D), although the lesser pathway of Aub-generated, Piwi trailing piRNAs is not affected (Fig 2D). The profound collapse of Aub–Ago3 ping-pong in $aub^{RK}$ ovaries extends to all transposon classes (Fig 2E). Collectively, our findings show that Aub–NTRs are dispensable for primary piRNA biogenesis, which takes place on the cytoplasmic surface of mitochondria, as Aub piRNAs in $aub^{RK}$ are identical to those from $aub^{WT}$. However, $Aub^{RK}$ is unable to interact with Krimp and build the piRNA amplification complex in nuage that would recruit unloaded Ago3 to receive the products of transposon cleavage by Aub piRNAs. As a result, heterotypic Aub–Ago3 collapses, Ago3 enters the primary piRNA pathway (Fig S2B), and Piwi piRNA population is altered, as trailing piRNAs disappear.

## Methylation of Aub–NTRs is largely dispensable for piRNA amplification

By replacing Aub–NTRs with lysines, the RK mutant abolishes methylation but also changes the arginines. To examine in more detail the Aub–NTR methylation itself in piRNA biogenesis and amplification, we employed $csul^{RM50}$, a genetic loss of function mutant of *Drosophila* PRMT5 (5, 7), and two short hairpin (sh) RNA knockdown lines, $csul^{TRiP1}$ and $csul^{TRiP2}$, generated by the Transgenic RNAi Project (TRiP). Germline knockdown of Csul was accomplished by driving shRNA expression with the triple Gal4 germline driver, which led to complete loss of Aub sDMAs in these flies (Fig 3A). WBs of ovary extracts from *csul* knockdown flies show reduction of Aub, Ago3, and Tud proteins (Fig 3B), similar to what we have previously reported for $csul^{RM50}$ (5). Unlike $Aub^{RK}$, nonmethylated Aub is found in nuage of $csul^{i1}$, although at lesser amounts and forming a thinner and less granular perinuclear circle than methylated Aub; the same is true for Krimp, Ago3, and Tud (Figs 3C and S3). To characterize the impact of nonmethylated Aub in piRNA biogenesis and amplification, we sequenced and analyzed Aub-bound and Ago3–piRNAs from $csul^{RM50}$ ovaries and compared them with those from *w* ovaries, expressing wild-type methylated Aub. We find that Aub piRNAs from $csul^{RM50}$ ovaries display 1U preference and Ago3 piRNAs show a 10A bias, similar to those from *w* ovaries (Fig 3D). The 5′-5′ distance between Aub and Ago3 piRNAs in $csul^{RM50}$ shows a peak at position 10 (red line), which is similar to that seen in *w* (Fig 3E) and similar Aub–Ago3 ping-pong z-scores for the various transposon classes (Fig 3F). These results indicate that nonmethylated Aub is still capable of assembling the piRNA amplification complex and engages in heterotypic Aub–Ago3 ping-pong for transposon control.

## Neither $Aub^{RK}$ nor nonmethylated Aub can assemble germplasm resulting in sterile offspring

Aub and Tud are essential components of the mRNP granules that constitute the germplasm (43), which by IF appears as a thick crescent at the oocyte posterior. In the presence of endogenous

indicated genotypes. $y w = y^1 w^1$, $aub = aub^{HN2/QC42}$. Tub serves as loading control. **(D)** Color-inverted confocal images depicting the localization pattern of indicated proteins (grey) in stage 4–7 egg chambers from indicated genotypes. Scale bar = 5 μm.
Source data are available for this figure.

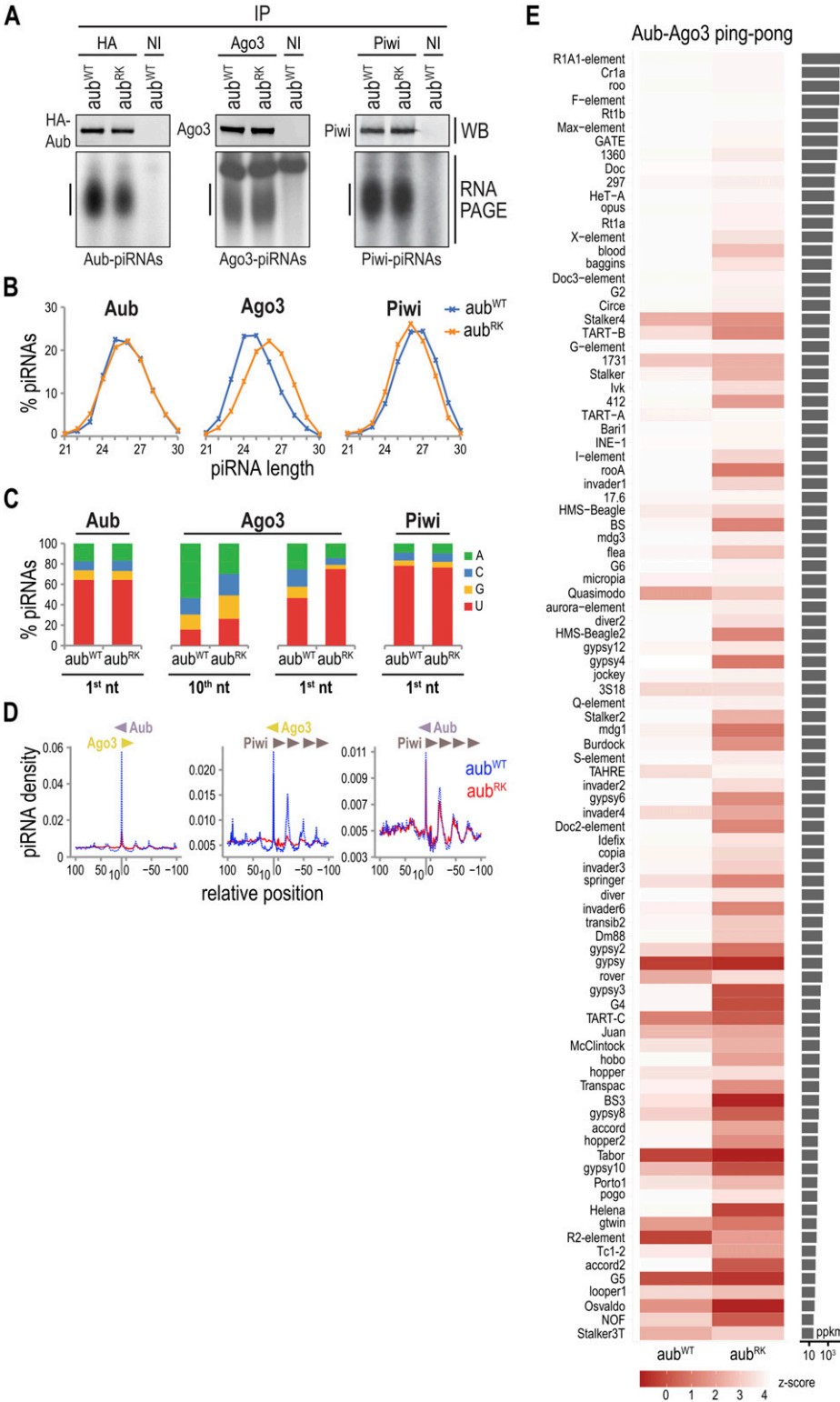

**Figure 2. Intact primary piRNA biogenesis and Aub[RK] piRNA loading but collapse of heterotypic ping-pong in aub[RK].**
**(A)** Proteins (top) and bound piRNAs (bottom) of immunoprecipitated Aub, Ago3, and Piwi from indicated genotypes. $aub^{WT,RK} = aub^{QC42/HN2}$; $nos > 3xHA-aub^{WT,RK}$. NI, nonimmune serum. **(B)** piRNA length distribution. **(C)** piRNA nucleotide composition. **(D)** Relative position of piRNA 5′ ends bound to indicated proteins. **(E)** Heat map representing z-scores for a 10-nt overlap between Aub–Ago3 transposon aligning piRNA pairs in indicated libraries. piRNA transposons are ranked by mean total piRNA abundance. ppkm, piRNA pairs per kilobase per million.
Source data are available for this figure.

Aub, Aub[RK] localizes to the germplasm (Fig S4A), indicating that the RNA binding of Aub[RK] is sufficient to recruit it to germplasm nucleated by endogenous Aub. In contrast, we find a drastic reduction of Aub and Tud at the oocyte posterior of stage 10 egg chambers in ovaries from $aub^{RK}$ (which lack endogenous Aub) and $csul^{i1}$, similar to that of $tud$ (Fig 4A). As a consequence, PGCs are not induced and the viable offspring of mothers expressing Aub[RK] or nonmethylated Aub never form a germline (Fig 4B).

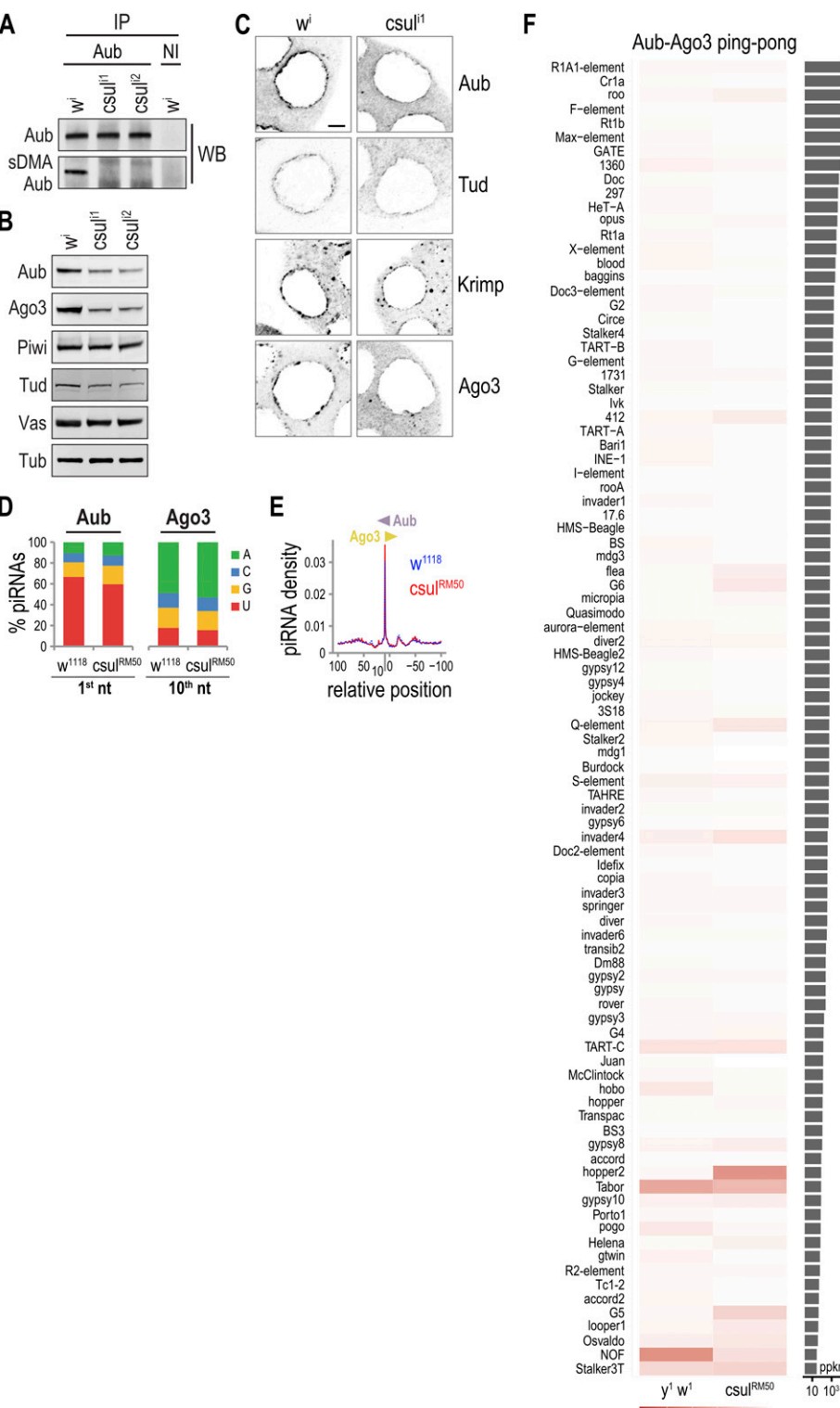

**Figure 3. piRNA biogenesis and ping-pong are largely intact in the absence of sDMAs.**
(A) Western blot detection of immunoprecipitated Aub from ovary lysates of indicated genotypes. $w^i$ = $MTD > w^{TRiP}$, $csul^{i1}$ = $MTD > csul^{TRiP1}$, $csul^{i2}$ = $MTD > csul^{TRiP2}$. NI, nonimmune serum, sDMA-Aub is detected with SYM11 antibody. (B) Western blot detection analysis in ovary lysates from indicated genotypes. Tub serves as loading control. (C) Color-inverted confocal images depicting the localization pattern of indicated proteins (grey) in stage 5–8 egg chambers from indicated genotypes. Scale bar = 5 $\mu$m. (D) piRNA nucleotide composition. (E) Relative position of piRNA 5′ ends bound to indicated proteins. (F) Heat map representing z-scores for a 10-nt overlap between Aub–Ago3 transposon-aligning piRNA pairs in indicated libraries. piRNA transposons are ranked by mean total piRNA abundance. ppkm, piRNA pairs per kilobase per million.
Source data are available for this figure.

## Discussion

Altogether our findings elucidate the role of Aub–NTRs and their methylation in transposon control and germplasm formation, in vivo. By replacing Aub–NTRs with lysines, essential interactions between Aub and Krimp and Aub and Tud are abolished, leading to collapse of piRNA amplification and transposon control, and of germplasm and germ cell specification, respectively. Although

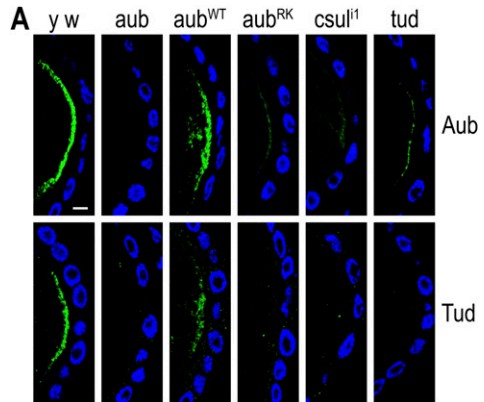

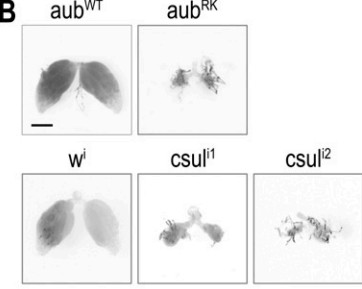

**Figure 4. Germplasm does not form when Aub lacks sDMAs or when arginines are replaced with lysines, resulting in sterile offspring.**
**(A)** Confocal images depicting Aub and Tud localization pattern (green) at the posterior pole of stage 10 egg chambers from indicated genotypes. Nuclei are stained with DAPI (blue). $y w = y^1 w^1$, $aub = aub^{QC42/HN2}$, $aub^{WT,RK} = aub^{QC42/HN2}$; $nos > 3xHA-aub^{WT,RK}$, $csul^{i1} = MTD > csul^{TRiP1}$, $tud = tud^1/Df(2R)Pu^{rP133}$. Scale bar = 10 μm. **(B)** Ovaries dissected from female adult offspring of indicated maternal genotypes. $csul^{i2} = MTD > csul^{TRiP2}$. Scale bar = 100 μm.

secondary piRNAs may induce the generation of additional primary piRNAs, our data align with previous reports (44, 45) that the amplification loop is not required for primary piRNA biogenesis. In addition, we provide in vivo evidence that the phased nature of primary piRNA processing may be decoupled from ping-pong, as previously suggested (39, 46).

$aub^{RK}$ shows high embryo lethality because transposons are deregulated. The small percentage of embryos that achieve adulthood most likely represent escapers, where transposon overexpression did not reach a lethal threshold during oocyte nucleus maturation. The few embryos that survive give rise to agametic offspring because germplasm does not form and germ cells are not specified. The hypomorphic character of $aub^{RK}$ allows us to dissect its pleiotropic role in the piRNA pathway and unmasks the grand-childless nature of the phenotype. Although this article was in the final stages of preparation, a preprint from the Aravin Lab (47 *Preprint*) showed a similar impact of Aub–RK mutant protein in piRNA amplification and elucidated the structural determinants of Krimp that build the piRNA amplification complex (47 *Preprint*).

By removing the methylation marks deposited by Csul in Aub–NTRs, we find that heterotypic ping-pong is largely intact, indicating that nonmethylated arginines are sufficient for interacting with Krimp, and likely other nuage TDRDs, to suppress transposons resulting in much higher embryo viabilities and normal somatic development of the offspring (6, 7). Aub regulates mRNAs in the embryo soma (48), and this function appears independent of Aub methylation and Aub–Tud interaction, given the viability and normal somatic development of *csul* and *tud* offspring. However, because nonmethylated Aub is unable to interact with Tud, germplasm does not form and offspring are agametic (Fig 4B). The biological significance of Aub–Tud interactions in germ granule assembly is further supported by the similar grand-childless phenotype of Tud loss of function. Somatic development in the absence of Tud still takes place, but germ granule mRNPs do not assemble, and PGCs are not induced (31). Among TDRDs, Tud has the largest number of eTUD domains that interact with sDMAs of Aub (33, 49). Along with Aub–piRNA binding of mRNAs (35), these multivalent interactions are critical for germ granule mRNP assembly. Our findings indicate that methylation of Aub–NTRs functions primarily to build germline mRNPs and may represent an evolutionary conserved pathway of germline mRNP formation. Notably, mammalian homologs of Aub and Tud, such as mouse Miwi (Piwil1) and mouse Tdrd6, are essential components of chromatoid bodies, which assemble in pachytene spermatocytes and round spermatids and are enriched in mRNAs (50). Miwi interaction with Tdrd6 is dependent on sDMAs of Miwi-NTR (9). It will be interesting to further explore the biological significance of such interaction in germline mRNP assembly in mammals.

# Materials and Methods

### Plasmid construction

WT and RK versions of *aub* were amplified with *PfuUltra* (Agilent) using previously published laboratory constructs as template (9) and the following primers CACCAATTTACCACCAAACCCTGTAAT and TTACAAAAA GTACAATTGATTCTGC. Amplicons were directionally cloned into pENTR/D-TOPO (Thermo Fisher Scientific) and recombined into Gateway vector pPHW (*Drosophila* Genomics Research Center). P-element–based *Drosophila* transgenesis followed (Genetic Services, Inc.).

### Fly husbandry

Flies were grown on standard cornmeal molasses at 25°C, with 70% relative humidity under a 12-h light–dark cycle. Virgin female flies were mated with *y w* males inside vials supplied with dry yeast for at least 2 d before downstream processing. A full list of lines used in this work is detailed in Table S3.

### Ovary immunofluorescence and confocal microscopy

Ovaries were dissected from 2- to 5-d-old flies inside cold Ringer's solution (10 mM HEPES pH 6.9 with KOH, 130 mM NaCl, 4.7 mM KCl, 1.9 mM $CaCl_2$). All wash and incubation steps required constant shacking in vertically placed tubes and were executed at room temperature, unless otherwise stated.

Grk, Tud, and Osk staining was based on reference 51 with minor modifications. Briefly, ovarioles were separated with fine forceps and fixed for 10 min. The fixative solution contained five volumes of n-heptane with 1 volume of devitalizing buffer (6.16% paraformaldehyde, 16.7 mM $KH_2PO_4$/$K_2HPO_4$ pH 6.8, 75 mM KCl, 25 mM NaCl, 3.3 mM $MgCl_2$). After three rinses in PBS (10 mM $Na_2HPO_4$, 1.8 mM $KH_2PO_4$ pH 7.4 with HCl, 137 mM $NaCl_2$, 2.7 mM KCl) and two more in $PT_3$ (0.3% Triton X in PBS), the material was blocked in $PBT_3$ (1% bovine serum albumin, 0.3% Triton X in PBS) for 2 h. Ovaries were incubated overnight at 4°C with appropriately diluted primary antibodies in $PBT_3$ (Table S4). The next day, after three washes in $PT_3$ for 30 min each, Alexa Fluor secondary antibody incubation followed for 2 h. Ovaries were subsequently washed three times in $PT_3$ for 30 min, rinsed twice in PBS, and incubated with DAPI staining solution (1 $\mu$M DAPI in PBS) for 10 min. After two PBS washes for 10 min each, the material was mounted with ProLong Gold (Thermo Fisher Scientific) and stored in the dark.

Aub, HA, Krimp, Ago3, Qin, Vas, and Piwi staining was adopted from reference 52 with minor modifications. Whole mount ovaries were fixed strictly for 5 min. Fixative solution is detailed in the previous paragraph. Ovaries were rinsed three times in PBS followed by three rinses in PT (0.1% Triton X in PBS) and 1-h incubation in the same buffer. Next, ovarioles were separated with fine forceps and further incubated for 1 h in PT. The material was blocked in PBT (1% BSA, 0.1% Triton X in PBS) for 2 h and then incubated overnight at 4°C with primary antibodies diluted in PBT (Table S4). The next day, after eight washes in PBT for 15 min each, Alexa Fluor secondary antibody incubation proceeded overnight at 4°C. The third day, ovaries were washed eight times in PT for 15 min each, rinsed twice in PBS, and further processed with DAPI staining and mounted exactly as described in the paragraph above.

Preparations were imaged on the Leica TCS and illustrated as single Z-stacks. Each protein was studied under identical microscope settings to permit comparison of signal intensity between genotypes.

### Lysate preparation, Western blot, and antibodies

Ovaries from 2- to 5-d-old yeast fed flies were dissected in cold PBS and pooled in batches of 50. The dissected material was flash-frozen in liquid nitrogen and stored at −80°C. For lysate preparation and WBs, ovaries were processed, as previously described (46), with the addition of TCEP to 0.5 mM in RSB-200 buffer. Ago3-380 antibody was produced by immunizing rabbits with synthetic peptide IKKSRGIPAERENL conjugated to KLH via an amino-terminal cysteine, followed by affinity purification of sera over columns containing the immobilized peptide (Genscript). Ago3-380 successfully detected and immunoprecipitated Ago3 protein in ovary lysates (Fig S4B). Antibodies used for WBs are listed in Table S5.

### Immunoprecipitation and RNA isolation

100 ovaries per sample were used in Aub and Piwi IP experiments and processed, as previously described (46). For Ago3 IP experiments, we used 150 ovaries per sample with a slight modification in the protocol. Ovary lysates were first incubated with 4 $\mu$g Ago3-380 antibody for 2 h at 4°C and then mixed with buffer-equilibrated Protein G Dynabeads (Thermo Fisher Scientific) for 90 min at 4°C. The RNAs associated with immunopurified PIWI proteins were extracted with TRIzol reagent (Ambion) and dephosphorylated with Quick CIP (NEB) in CutSmart buffer for 10 min at 37°C. After enzyme inactivation for 2 min at 80°C, a T4 PNK (NEB) labeling reaction was set in 1× CutSmart buffer with the addition of DTT to 5 mM in the presence of $^{\gamma 32P}$ATP. Reactions were run with 8 M urea 15% PAGE.

### Small RNA library construction

Aub, Ago3, and Piwi piRNA libraries from *y w*, $aub^{WT}$, and $aub^{RK}$ were constructed, as described in reference 46. Sequence information from a previously published *y w* ovarian Aub-IP library was retrieved from reference 35. Aub and Ago3 piRNAs from $w^{1118}$ and $csul^{RM50}$ were isolated and processed into libraries, as detailed in reference 5. A complete list of the libraries produced for this work is detailed in Table S6.

### Read processing, alignment, and computational analyses

The 3′ end adaptor sequence was trimmed from all reads using Cutadapt with parameters -m 15 -e 0.25. For libraries with 8-nt random barcode at the 3′ end, an additional sequence collapsing step was performed to discard PCR duplicates. In that step, identical reads were collapsed, and only one was retained using CLIPSeqTools (53). Afterward, the 8-nt barcode was removed. Reads were aligned to the *Drosophila melanogaster* genome (dm3) using STAR v2.4.2 using the following parameters: outFilterMultimapScoreRange 0, alignIntronMax 50000, outFilterIntronMotifs, RemoveNoncanonicalUnannotated, outFilterMatchNmin 15, outFilterMatch, NminOverLread 0.9, and sjdbOverhang 50. The reference gene model annotation file was downloaded from the University of California Santa Cruz (UCSC) genome browser database. Aligned reads were loaded into an SQLite3 database for further processing with CLIPSeqTools and were annotated based on whether they were contained in elements from RepeatMasker (downloaded from UCSC), ribosomal RNAs (extracted from UCSC gene model annotation), transfer RNAs (downloaded from FlyBase r5.57), piRNA clusters, and genes (from UCSC gene model annotation file). Reads were also aligned to consensus transposon sequences using STAR with the following parameters: outFilterMultimapScoreRange 0, alignIntronMax 1, alignEndsType EndToEnd, seedSearchStartLmax 20, outFilterMatchNmin 15, and outFilterMatchNminOverLread 0.95. The consensus sequences for transposable elements (v9.42) were downloaded from FlyBase. For ping-pong analysis, the relative position distribution for Aub–Ago3 transposon aligning piRNA pairs was calculated. Density values

for all positions were converted to standard scores (z-scores). Heat maps demonstrate the z-score for the 10-nt overlap. piRNA transposons were ranked by mean total piRNA abundance.

## Data Availability

Sequencing data have been deposited into the Sequence Read Archive, project ID: GSE155874.

## Supplementary Information

## Acknowledgements

We would like to thank M Siomi, T Kai, J Brennecke, and the Developmental Studies Hybridoma Bank for antibodies. We are grateful to Ch Delidakis, T Jongens, N Perrimon, R Binari, J Anne, A Arkov, the *Drosophila* Bloomington Stock Center, and the *Drosophila* Genomics Research Center for fly lines and the Gateway vector. We thank Jonathan Schug at UPenn Next-Generation Sequencing Core for high-throughput sequencing. Z Mourelatos and laboratory are supported by the National Institutes of Health (NIH) grant GM123512. M Maragkakis is supported by the Intramural Research Program of the National Institute on Aging, NIH. P Alexiou is supported by the Czech Science Foundation (GACR) grant 19-10976Y. Y Kirino is supported by NIH grant GM106047.

### Author Contributions

N Vrettos: conceptualization, data curation, formal analysis, validation, investigation, visualization, methodology, and writing—original draft, review, and editing.
M Maragkakis: data curation, software, formal analysis, validation, investigation, visualization, methodology, and writing—original draft, review, and editing.
P Alexiou: software, formal analysis, and writing—review and editing.
P Sgourdou: investigation, visualization, methodology, and writing—review and editing.
F Ibrahim: investigation, visualization, methodology, and writing—review and editing.
D Palmieri: validation and methodology.
Y Kirino: resources, investigation, methodology, and writing—review and editing.
Z Mourelatos: conceptualization, resources, formal analysis, supervision, funding acquisition, investigation, project administration, and writing—original draft, review, and editing.

### Conflict of Interest Statement

The authors declare that they have no conflict of interest.

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
