## [Reviewer comments · Life Science Alliance]

Life Science Alliance

Modulation of Aub - TDRD interactions elucidates piRNA amplification and germ plasm formation

Nicholas Vrettos, Manolis Maragkakis, Panagiotis Alexiou, Paraskevi Sgourdou, Fadia Ibrahim, Daniel Palmieri, Yohei Kirino, and Zissimos Mourelatos

DOI: <https://doi.org/10.26508/lsa.202000912>

Corresponding author(s): Zissimos Mourelatos, University of Pennsylvania

Review Timeline:

Submission Date:	2020-09-22
Editorial Decision:	2020-11-19
Revision Received:	2020-12-13
Accepted:	2020-12-14

Scientific Editor: Shachi Bhatt

Transaction Report:

November 18, 2020

RE: Life Science Alliance Manuscript #LSA-2020-00912-T

Dr Zissimos Mourelatos
Pennsylvania, University of
University of Pennsylvania School of
Medicine
613B Stellar Chance Labs
Philadelphia, 422 Curie Boulevard USA-Philadelphia, PA 19104-6100

Dear Dr. Mourelatos,

Thank you for submitting your revised manuscript entitled "Modulation of Aub - TDRD interactions elucidates piRNA amplification and germ plasm formation". We would be happy to publish your paper in Life Science Alliance pending final revisions necessary to meet our formatting guidelines.

As you will note from the reviewer comments below, the reviewers found the data quite intriguing and are highly supportive of publishing the study in Life Science Alliance, pending minor text edits.

In addition to the reviewers' requests and the points listed below, please also attend to the following:

- please consult our Manuscript Preparation Guidelines <https://www.life-science-alliance.org/manuscript-prep> and put your manuscript sections in the correct order
- please add Author Contributions to the system
- please add a Summary Blurb / Alternate Abstract to the system
- please add a conflict of interest statement in the system and in your main manuscript text
- please upload your main and supplementary figures as single files
- please upload your main manuscript text as an editable doc file
- please upload your tables as editable doc or excel files and add your table legends to the main manuscript text
- please clarify the Category for the manuscript for publication
- please add a scale bar to Fig 4B
- please move the RNA seq accession info to a separate section labeled Data Availability (<https://www.life-science-alliance.org/manuscript-prep#datadepot>)
- please provide a point-by-point response to all the concerns raised by the reviewers

To avoid unnecessary delays in the acceptance and publication of your paper, please read the

following information carefully.

A. FINAL FILES:

B. MANUSCRIPT ORGANIZATION AND FORMATTING:

Sincerely,

Shachi Bhatt, Ph.D.
Executive Editor
Life Science Alliance
<https://www.lsjournal.org/>
Tweet @SciBhatt @LSAJournal

Reviewer #1 (Comments to the Authors (Required)):

In their manuscript "Modulation of Aub - TDRD interactions elucidates piRNA amplification and germ plasm formation", Vrettos et al. dissect the functions of the Drosophila PIWI protein Aubergine (Aub) in (a) secondary piRNA production during ping-pong, and (b) the formation of germ plasm. The authors identify a differential role for methylated and non-methylated (Aub-NTR) Aub through elegant genetic and biochemical experiments. Their results show that methylation is required for ping-pong amplification of piRNAs and for germ plasm, which induces germ cells in the next generation. However, methylation was dispensable for phased piRNA biogenesis and the function of Aub-piRNA complexes in transposon restriction in germ cells. Accordingly, aub-NTR flies are fertile but exhibit a grandchildless phenotype. Through precise and careful molecular experiments the authors can overcome the sterility phenotype of aub and observe its grandchildlessness for the first time. The presented data greatly improve our mechanistic understanding of piRNA biogenesis and reveal an independent function for Aub in germ plasm. I highly recommend this manuscript for publication with minor revisions.

Here, the authors show a function for maternally contributed PIWI-piRNA complexes in vivo for the first time. In previous studies this phenotype was concealed by the sterility of all PIWI null mutants. The authors might want to emphasize this unique view into the fully mutant F1 and the lack of the F2 generation. These data can explain the differential phenotypes of aub, the methyltransferase (capsuleen) and the two tudor proteins, krimper and tudor.

The authors show that primary PIWI-piRNA complexes are sufficient for fertility. The finding that ping-pong is (largely) dispensable for the function of piRNA-guided genome defense in healthy animals could be stated more explicitly, and the authors could speculate about a potential requirement for ping-pong under exogenous or endogenous stress situations.

Text page 2: "PIWI proteins are intimately involved in piRNA biogenesis using their MID domain to bind the 5' phosphate of longer piRNA precursors, ..." The authors should clarify that while secondary piRNAs have been suggested to induce additional primary piRNA biogenesis, they are not required for primary piRNA biogenesis (Saito et al., Nature 2009; Sumiyoshi et al., Genes Dev. 2016). Clarifying the partial nature of previous data would help the reader to understand that there is no contradiction between the authors novel results and previous reports. The authors elegantly show that the phased appearance of primary piRNAs is independent of ping-pong processing, which has been suggested previously in a tissue culture system (Stein et al., 2019). The authors could speculate on the processive or distributive nature of primary processing in the discussion.

Figure 1: no comments

Figure 2: panel A: It is surprising that in the absence of methylated Aub, Ago3 seems to be loaded with primary piRNAs. The authors should show an annotation of Aub, Ago3 and Piwi-piRNAs with respect to their targeting potential: as sense or antisense transposon (transposon families). In panel C, the authors could change the colors to agree with the general color-code: A=green, C=blue, G=yellow, U=red. This could help the reader's orientation. (same in Fig. 3D)
Panel E, the authors should clarify the unit ppkm in the figure legend.

Figure 3: no comments

Figure 4: panel B: The authors should show the ovaries of the parents and of the heterozygous grandparents for comparison and to emphasize that we are looking at grandchild ovaries of a piRNA pathway mutant for the first time.

Reviewer #2 (Comments to the Authors (Required)):

Piwi-interacting RNAs (piRNAs) repress transposable elements in animal germlines. The *Drosophila* Piwi protein Aub, is post-translationally modified by symmetrical dimethyl arginine modifications (sDMA) on the N-terminus. These are recognized by Tudor domains in Tudor-domain containing (TDRD) proteins. These proteins along with many others form a piRNA biogenesis complex that facilitates the so-called Ping-pong piRNA biogenesis where Aug cleavage of a target results in loading of piRNAs into another Piwi protein Ago3, and also leads to series of phased piRNAs. In addition to piRNA biogenesis, Piwi methylation and Tudor proteins have a role in formation of the germ plasm in the oocyte, which mediates the formation of the future germline in the developing embryo.

The authors prepare a fly line which expresses a version of Aub which lacks the N-terminal arginines (which were converted to lysines; *aubRK*). By analysing this fly line, they demonstrate that these arginines are required for proper development of the laid eggs into adults, and for formation of the germline. Localization of the various players in the pathway is affected and fail to form structures called nuage in the cytoplasm where a particular type of the piRNA biogenesis takes place. Indeed, they show that a branch of the piRNA biogenesis called Ping-pong cycle is abolished in this mutant. The consequence of this is loss of the downstream phased piRNAs. They find that methylation of the arginines in Aub is not required for piRNA biogenesis, but is essential for germ

plasm formation and fertility.

The findings are clear and supported by strong evidence. The manuscript is of interest to the small RNA and germ cell biology community. I support its immediate publication.

Comments

Page 4: "receptor protein Tud". The wording may be re-considered. "Receptor" maybe misunderstood by those outside the field.

Page 7. "Ago3 piRNAs in aubRK are longer by one nucleotide". It is worth mentioning here or later that the same observation is made in Tudor mutants.

Figure 2D: Details on the phasing distances can be provided in the main text and in the figure.

Figure 3C: Provide scale bars. Some of the nuclei look smaller. Is the scale same for all?

Reviewer #3 (Comments to the Authors (Required)):

In the manuscript the authors have successfully showed that arginines at the N-terminus of Aub (Aub-NTRs) play a crucial role in piRNA biogenesis in *Drosophila melanogaster* and specifically in piRNA amplification, while their symmetric demethylation is required for germ plasm formation. The authors engineered an aub mutant by replacing four arginine residues with lysines (RK) and inserting 3 tandem HA epitopes at the N-terminus. By comparing aub mutant with aub wild type ovaries they determined that (i) even if the protein levels involved in piRNA biogenesis pathway are not disrupted, females from aub mutants exhibit low fecundity rates and carry eggs with severe axonal defects, (ii) in Aub absence, nuage structures are not built, while Tud and Krimp are absent from nuage, (iii) a drastic reduction of ping-pong amplification is noticed and the viable offspring of mothers expressing aub mutant or non-methylated Aub never form a germ line.

The authors' findings are very important and provide new insights regarding piRNA biogenesis pathway in flies. They present clearly all their results and the data they produced were fully supportive to their findings. In my view the paper is ready for publication and will valuably contribute to the field.

Thank you very much for positively evaluating our work. We would like to thank the reviewers for assessing our study and for their insightful comments and suggestions. Point by point we demonstrate the changes we incorporated in the manuscript and hope that the updated version will be appropriate for publication in Life Science Alliance.

We reformatted the whole manuscript according to the preparation guidelines. The manuscript sections are in the correct order and we added: Author contributions, a summary blurb, a conflict of interest statement and a data availability section.

We consider our study a research article focusing in small RNA regulation of Drosophila development and stem cell biology.

We added a **scale** bar on Fig 4B, we thank you and the reviewer #1 for pointing out this oversight.

We added **3** pdf files with the raw source data for each of our main figures.

We gladly accept for all review, decision and response process to be published online alongside our paper.

Reviewer #1

In their manuscript "Modulation of Aub - TDRD interactions elucidates piRNA amplification and germ plasm formation", Vrettos et al. dissect the functions of the Drosophila PIWI protein Aubergine (Aub) in (a) secondary piRNA production during ping-pong, and (b) the formation of germ plasm. The authors identify a differential role for methylated and non-methylated (Aub-NTR) Aub through elegant genetic and biochemical experiments. Their results show that methylation is required for ping-pong amplification of piRNAs and for germ plasm, which induces germ cells in the next generation. However, methylation was dispensable for phased piRNA biogenesis and the function of Aub-piRNA complexes in transposon restriction in germ cells. Accordingly, aub-NTR flies are fertile but exhibit a grandchildless phenotype. Through precise and careful molecular experiments the authors can overcome the sterility phenotype of aub and observe its grandchildlessness for the first time. The presented data greatly improve our mechanistic understanding of piRNA biogenesis and reveal an independent function for Aub in germ plasm. I highly recommend this manuscript for publication with minor revisions.

Here, the authors show a function for maternally contributed PIWI-piRNA complexes in vivo for the first time. In previous studies this phenotype was concealed by the sterility of all PIWI null mutants. The authors might want to emphasize this unique view into the fully mutant F1 and the lack of the F2 generation. These data can explain the differential phenotypes of aub, the methyltransferase (capsuleen) and the two tudor proteins, krimper and tudor.

The authors show that primary PIWI-piRNA complexes are sufficient for fertility. The finding that ping-pong is (largely) dispensable for the function of piRNA-guided genome defense in healthy animals could be stated more explicitly, and the authors could speculate about a potential requirement for ping-pong under exogenous or endogenous stress situations.

We thank the reviewer for appraising our findings. Indeed, the hypomorphic character of our mutant unmasks the grandchildless nature of Aub. We now highlight this matter in our manuscript. Although primary piRNA biogenesis pathway appears undisturbed in our mutant, the lack of a robust ping-pong amplification wave renders the genome exposed to active transposons, justifying the high offspring lethality. We strongly agree that the aggravated phenotype of our mutant would worsen, in the case where flies were stressed by an exogenous or endogenous stimulus like a newly introduced transposon strain.

Text page 2: "PIWI proteins are intimately involved in piRNA biogenesis using their MID domain to bind the 5' phosphate of longer piRNA precursors, ..." The authors should clarify that while secondary piRNAs have been suggested to induce additional primary piRNA biogenesis, they are not required for primary piRNA biogenesis (Saito et al., Nature 2009; Sumiyoshi et al., Genes Dev. 2016). Clarifying the partial nature of previous data would help the reader to understand that there is no contradiction between the authors novel results and previous reports.

The authors elegantly show that the phased appearance of primary piRNAs is independent of ping-pong processing, which has been suggested previously in a tissue culture system (Stein et al., 2019). The authors could speculate on the processive or distributive nature of primary processing in the discussion.

We thank the reviewer for indicating a connection between our findings and previous work. We incorporated the information and citations the reviewer is suggesting in our manuscript.

Figure 1: no comments

Figure 2: panel A: It is surprising that in the absence of methylated Aub, Ago3 seems to be loaded with primary piRNAs. The authors should show an annotation of Aub, Ago3 and Piwi-piRNAs with respect to their targeting potential: as sense or antisense transposon (transposon families).

We appreciate the reviewer's recommendation. Accordingly we prepared a genome browser view of how sense and antisense piRNAs map to representative loci. This illustration has been incorporated in our manuscript as panel B in Supplementary Figure 2.

authors could change the colors to agree with the general color-code: A=green, C=blue, G=yellow, U=red. This could help the reader's orientation. (same in Fig. 3D)

We updated both figures 2 and 3 with the suggested color rule.

Panel E, the authors should clarify the unit ppkm in the figure legend.

We thank the reviewer for highlighting this omission. We updated the figure legend with the right information.

Figure 3: no comments

Figure 4: panel B: The authors should show the ovaries of the parents and of the heterozygous grandparents for comparison and to emphasize that we are looking at grandchild ovaries of a piRNA pathway mutant for the first time.

Following the reviewer's suggestion we emphasized the grandchildless nature of our mutant. Yet we feel it would be redundant to document the presence or absence of germ line in the parental and filial generations.

Reviewer #2

Piwi-interacting RNAs (piRNAs) repress transposable elements in animal germlines. The Drosophila Piwi protein Aub, is post-translationally modified by symmetrical dimethyl arginine modifications (sDMA) on the N-terminus. These are recognized by Tudor domains in Tudor-domain containing (TDRD) proteins. These proteins along with many others form a piRNA biogenesis complex that facilitates the so-called Ping-pong piRNA biogenesis where Aug cleavage of a target results in loading of piRNAs into another Piwi protein Ago3, and also leads to series of phased piRNAs. In addition to piRNA biogenesis, Piwi methylation and Tudor proteins have a role in formation of the germ plasm in the oocyte, which mediates the formation of the future germline in the developing embryo.

The authors prepare a fly line which expresses a version of Aub which lacks the N-terminal arginines (which were converted to lysines; aubRK). By analysing this fly line, they demonstrate that these arginines are required for proper development of the laid eggs into adults, and for formation of the germline. Localization of the various players in the pathway is affected and fail to form structures called nuage in the cytoplasm where a particular type of the piRNA biogenesis takes place. Indeed, they show that a branch of the piRNA biogenesis called Ping-pong cycle is abolished in this mutant. The

consequence of this is loss of the downstream phased piRNAs. They find that methylation of the arginines in Aub is not required for piRNA biogenesis, but is essential for germ plasm formation and fertility.

The findings are clear and supported by strong evidence. The manuscript is of interest to the small RNA and germ cell biology community. I support its immediate publication.

Comments

Page 4: "receptor protein Tud". The wording may be re-considered. "Receptor" maybe misunderstood by those outside the field.

This is a very valid point. We have accordingly deleted this term in the text.

Page 7. "Ago3 piRNAs in aubRK are longer by one nucleotide". It is worth mentioning here or later that the same observation is made in Tudor mutants.

We thank the reviewer for pointing out this analogy. We have incorporated this information in our manuscript with the appropriate citations.

Figure 2D: Details on the phasing distances can be provided in the main text and in the figure.

Following the reviewer's suggestion, we have included information about phasing distance in our manuscript. We believe it is redundant to add the length distance on figure 2D since x axis is graded.

Figure 3C: Provide scale bars. Some of the nuclei look smaller. Is the scale same for all?

We thank the reviewer for bringing up this matter. In our work every IF experiment is demonstrated by a series of panels placed horizontally with each protein target detected, indicated on the right. Each subfigure legend specifies the range of egg-chamber stages that were captured. We consistently use the same egg-chamber stage for documenting each protein's localization pattern in different genotypes. The stage of oogenesis varies between different IF protein targets but it never does in the same experiment. Microscope, zoom and objective settings are all identical. This is the reason for which we employ one scale bar per subfigure.

Reviewer #3

In the manuscript the authors have successfully showed that arginines at the N-terminus of Aub (Aub-NTRs) play a crucial role in piRNA biogenesis in *Drosophila melanogaster* and specifically in piRNA amplification, while their symmetric demethylation is required for germ plasm formation. The authors engineered an aub mutant by replacing four arginine residues with lysines (RK) and inserting 3 tandem HA epitopes at the N-terminus. By comparing aub mutant with aub wild type ovaries they determined that (i) even if the protein levels involved in piRNA biogenesis pathway are not disrupted, females from aub mutants exhibit low fecundity rates and carry eggs with severe axonal defects, (ii) in Aub absence, nuage structures are not built, while Tud and Krimp are absent from nuage, (iii) a drastic reduction of ping-pong amplification is noticed and the viable offspring of mothers expressing aub mutant or non-methylated Aub never form a germ line.

The authors' findings are very important and provide new insights regarding piRNA biogenesis pathway in flies. They present clearly all their results and the data they produced were fully supportive to their findings. In my view the paper is ready for publication and will valuably contribute to the field.

We thank the reviewer for such positive appraisal of our work.

Thank you very much for considering publishing our study in Life Science Alliance.

Sincerely

Zissimos Mourelatos

December 14, 2020

RE: Life Science Alliance Manuscript #LSA-2020-00912-TR

Zissimos Mourelatos
Pennsylvania, University of
University of Pennsylvania School of
Medicine
613B Stellar Chance Labs
Philadelphia, 422 Curie Boulevard USA-Philadelphia, PA 19104-6100

Dear Dr. Mourelatos,

Thank you for submitting your Research Article entitled "Modulation of Aub - TDRD interactions elucidates piRNA amplification and germ plasm formation". It is a pleasure to let you know that your manuscript is now accepted for publication in Life Science Alliance. Congratulations on this interesting work.

DISTRIBUTION OF MATERIALS:

Again, congratulations on a very nice paper. I hope you found the review process to be constructive and are pleased with how the manuscript was handled editorially. We look forward to future exciting submissions from your lab.

Sincerely,

Shachi Bhatt, Ph.D.

Executive Editor

Life Science Alliance

<https://www.lsjournal.org/>
